# A New Extension to a Multi-Criteria Decision-Making Model for Sustainable Supplier Selection under an Intuitionistic Fuzzy Environment

**Patchara Phochanikorn [1] and Chunqiao Tan [2,]***

[1] School of Business, Central South University, Changsha 410083, Hunan, China; 151608001@csu.edu.cn
[2] School of Business, Nanjing Audit University, Nanjing 211815, Jiangsu, China
[*] Correspondence: chungqiaot@mail.csu.edu.cn; Tel.: +86-731-88830594

**Abstract:** The increase of environmental pollution has led to the rise of sustainable awareness in recent years. This trend has motivated various industries to recognize the importance of implementing sustainable supply chain practices to seek economic, environmental and social advantages. From a sustainability perspective, selecting a suitable supplier is the main component of modern enterprises. It is also a challenging problem since several criteria concerning supplier selection are interdependent with a complex character. Therefore, the contribution of this paper is a new extension to multi-criteria decision-making model (MCDM) under an intuitionistic fuzzy environment for sustainable supplier selection (SSS) based on sustainable supply chain management SSCM practices. It consists of intuitionistic fuzzy set theory (IFS) with a decision making trial and evaluation laboratory (DEMATEL) combined with an analytic network process (ANP) to identify uncertainties and interdependencies among criteria as well as analyzing the criteria weights. We modified Vise Kriterijumska Optimizacija I Kompromisno Resenje (VIKOR) to evaluate and rank the desired level of sustainable supplier performance. The suggested approach is conducted by a case study from the Thailand palm oil industry. Results show that the proposed model not only can find the most suitable sustainable supplier, but also the enterprises can aid their suppliers in improving sustainability by using the proposed method and can improve enterprises' socio-environmental performance, which is key to achieving sustainable development.

**Keywords:** sustainable supplier selection; sustainable supply chain management; sustainable development; intuitionistic fuzzy set; multi-criteria decision-making

## 1. Introduction

In the past, various activities in the supply chain were conducted in a simple linear model composed of a small number of stakeholders. Therefore, most companies focused on enhancing the efficiency in economic dimensions such as technical quality development, cost reduction and product delivery speed. Later, the business became increasingly complex and now consists of more stakeholders [1]. As a result, the relationship between the activities in the supply chain has shifted from a linear model to a network model, and customer needs are the major driving force in determining the relationship model in the supply chain system, and whether suppliers, manufacturers and distributors need to improve the competitiveness through cooperation [2]. Besides, sustainable development in the modern business world has earned increasing attention. Essentially, in production operation management, the balancing of economic benefits and sustainable development has become a crucial topic for modern enterprises [3].

Sustainable supply chain management (SSCM) is one of the key factors for the success of the globalization business which is highly competitive under rapidly changing environments [4]. It is an

effective management model that includes economic, environmental, and social performances [5,6]. SSCM involves several practices, such as sustainable supplier selection, sustainable production and sustainable products [5,7,8]. In particular, sustainability goals can be achieved through the sourcing of suppliers for both focal companies and the entire supply chain [9–11].

Sustainable supplier selection (SSS) is a complicated multi-criteria decision-making and critical decision, which will impact the success or failure of supplier operation. Many studies use a variety of sustainability concept measurement criteria to assess appropriate suppliers. However, the criteria used in studies include performance criteria and practice criteria [5,12,13]. Scholars have further studied the relationship between SSCM practices and SSCM performances. In general, SSCM practices have a significant positive effect on the organization' performances [12,14]. Thus, utilizing management practices to access and choose suppliers is an easier way and more convenient. Applying practice criteria to supplier selection could help the enterprise to focus on target suppliers quickly, which is significant for early development of suppliers. Based on the review of existing research, it is concluded that the selection of green suppliers focuses on economic and environmental criteria [8,15,16], and there is still a lack of consideration of social performance practices in procurement operations [17]. At the same time, the inclusion of sustainability criteria into traditional supplier selection practices is also a requirement for supplier selection [18,19]. Therefore, how to establish an effective evaluation system and method mainly considering finding sustainable suppliers is a significant and challenging task for purchasing manager and researchers [20,21].

In real-life, human beings are always faced with the decision-making process under uncertain environments and fuzzy decision information. The traditional way which represents human thinking indicates data in the form of a crisp number, but it has many weaknesses. For example, the crisp number cannot handle the uncertainty of human judgment, which may lead to information loss and decrease the efficiency of decision-making analysis such as evaluating and alternative ranking. In order to overcome this deficiency, the linguistic number should be evaluated more conveniently and reliably. In the field of supplier selection method, most research uses MCDM techniques based on conventional fuzzy set theory to determine the weights and solve uncertain information without analyzing the impact of each factor on the final decision-making results [22,23]. Furthermore, the limitation of fuzzy set theory is that it is not sufficient for denoting the data on human judgments.

Based on the above discussion, this paper aims to strengthen the decision-making process and thoroughly analyze the supplier selection in uncertain environments. Based on the study results, enterprises can not only enhance their socio-environmental performance but also help their suppliers to improve sustainability, which is crucial to realize sustainable development. To achieve these purposes, this study makes the following improvements:

(1) Establish the criteria influencing SSS based on SSCM practices for supplier evaluation, which are grouped into the economic, environmental, and social aspects.

(2) Considering sustainable suppliers based on conflicting criteria and uncertain decision information, intuitionistic fuzzy set theory (IFS) with a decision making trial and evaluation laboratory (DEMATEL) is extended to analyze the cause and effect relationships. Moreover, an approach featuring a DEMATEL-based analytic network process (ANP) is used to calculate the substantial weight of the criteria [24]. This approach aims to overcome the dependence and feedback that accompany the selection criteria and other alternatives. Finally, from a set of alternatives in the supply chain, Vise Kriterijumska Optimizacija I Kompromisno Resenje (VIKOR) is modified to evaluate the sustainable supplier with the highest efficiency. The new ranking method is called IF-DANP-mV.

(3) To prove the feasibility of this method and to realize sustainable development by enhancing SSS based on SSCM practice, it is applied to a real case, in which a Thailand palm oil product industry intends to select sustainable suppliers.

This study has three main contributions. First of all, considering the economic, environmental, and social aspects in the SSCM condition, the paper establishes 3 dimensions and 13 criteria, which can help enterprises to identify the potential improvement areas for sustainable suppliers on the premise

of avoiding the potential risk of selecting unsuitable suppliers. Secondly, suppliers can apply the ranking result of relevant SSS criteria into their operations. Suppliers may enhance the long-term relationship with buyers by promoting their sustainable practices as valuable contributions to the sustainable supply chain. Finally, a new extension to the MCDM model is designed for SSS under an intuitionistic fuzzy environment to solve a problem in the selection of sustainable suppliers in an uncertain environment. Also, the findings of the study can improve management practices concerning the criteria to be used in SSS problems and provide an accurate sustainable supplier ranking and a reliable solution for sourcing decisions validated by a company.

The remainder of this paper is organized as follows. Section 2 presents a brief literature review on SSS based on SSCM practices. Section 3 presents a new extension of the MCDM model with IFS for SSS based on SSCM practices. Section 4 exemplifies a case study and discusses the results presented in Section 5. Finally, Section 6 concludes the paper.

## 2. Literature Review

### 2.1. SSS Based on SSCM Practices

Recently, there has been an increase in the global trends on environmental sustainability policy and practices [12,25–28]. The difference between a sustainable supply chain and green supply chain is that the basic goal of the green supply chain is to reduce waste and greenhouse gases in production. Yet, the definition of a green supply chain generally emphasizes the characteristics of environmental process flow [29], whereas, the scope of a sustainable supply chain is developed based on green supply chain management considering economic, environmental, and social dimensions [28]. Carter and Jennings [28] suggested that social awareness and environmental preservation are important components for an organization. Therefore, organizations should focus on environmental, social, and economic goals to achieve the organization strategy. Seuring and Müller [30] explained that SSCM is a strategic management that covers risk management, strategy and corporate based on the triple bottom line principle, including systematic coordination of business processes. Meanwhile, sustainability is receiving consumers' attention regarding the relationship between organizations and their suppliers in order to understand sustainable development [31]. Hence, social responsibility should be considered when investigating the sustainability of suppliers [17].

In recent reviews, some authors integrated sustainability into their organizations. For example, Paulraj et al. [32] established three dimensions, namely, sustainable product design, sustainable process design, and demand-side sustainability collaboration. Das [33] further incorporated three dimensions into the SSCM practices, including environmental management, social inclusiveness, operation performance, and supply chain integration. Meanwhile, Vargas et al. [14] studied organizational capabilities by focusing on the environmental and social aspects of SSCM practices.

Economic criteria aim to increase the profit flow that could be yielded while reducing the investment capital [34], such as, cost, quality, delivery, service, price of products, profit on products and flexibility [4,5,35,36]. The assessment of suppliers' environmental performance is defined from the environmental perspective, including green image, environmental management system, environmentally friendly materials, etc. [4,5,37,38]. From the social aspect, social management focuses on working place and employees' related determinants, such as protection of employees' right, and safe and healthy working environments [5,34,39]. Considering the reviews in all three dimensions of sustainability, most researchers used different criteria in their research. In the present study, the widely used conventional and sustainable criteria were combined for supplier assessment [20]. Moreover, the importance of each criterion was assessed and weighed using expert judgment (shown in Table 1).

**Table 1.** Sustainable supplier selection criteria.

| Dimension | Criteria | Description |
|---|---|---|
| Economic ($C_1$) | Product quality improvement ($C_{11}$)<br>Process capability ($C_{12}$)<br>Cost reduction activities ($C_{13}$)<br>Supply capability ($C_{14}$)<br>Delivery and service of product ($C_{15}$) | This section focuses on engaging the supply base in reference to the optimal means to reduce the cost and total price in terms of inputs/raw materials. For example, the lead time should include sufficient preparation and delivery times to enable the supplier to provide the purchasers with recycled materials. |
| Environmental ($C_2$) | Green image ($C_{21}$)<br>Environmental friendly materials ($C_{22}$)<br>Environmental policies ($C_{23}$)<br>Environmental planning ($C_{24}$)<br>Implementation and operation ($C_{25}$) | Suppliers and consumers should form organizations in which they can share environmental best management practices and knowledge in order to promote green initiatives, reduce environmental impacts, recycling, reduction, or elimination of all types of wastes, such as water and energy as well as ensure future development and cooperation. |
| Social ($C_3$) | Employer rights and welfare ($C_{31}$)<br>Safety and health system ($C_{32}$)<br>Stakeholder relationship ($C_{33}$) | By assessing the social aspect, the enterprises provide a safe and healthy policy for employer as well as increase workers' satisfaction such as wages, training, education. Moreover, the enterprise can also collaborate with stakeholders to improve the market shares which can enhance the image among the group of sector operators. |

*2.2. Sustainable Supplier Selection Methods*

This section provides reviews of the MCDM methods and reviews the combined IFS with MCDM methods regards to the SSS from earlier studies in the field related to the scope of this research.

2.2.1. MCDM Methods for SSS

Several methods for supplier selection, from a basic single method to complex multi-objective methods, have been developed and proposed [40]. In order to determine the best approach in solving problems related to decision making, researchers have used different approaches based on MCDM [5]. By combining more than two techniques, hybrid methods have recently received increasing attention due to their flexibility [41]. To position our study in this literature set, we aim to review popular MCDM methods that have been adopted in previous studies on sustainable supplier selection.

To explore and rank the important measures for sustainable supplier selection, Gören [34] presented a sustainable supply chain decision framework for an online retailer company. The model calculated the performance value of each supplier by using DEMATEL and the Taguchi loss functions, in order to determine the weights of the dependent criteria. Results showed that suppliers' ranking differed with the common MCDM methods. Hamurcu and Eren [42] combined ANP and Technique for Order Performance by Similarity to Ideal Solution (TOPSIS) methods, which provided a suitable ranking of Monorail routes. Based on the combination of MCDM model and best-worst method (BMW), Liu et al. obtained the optimal weights of sustainable suppliers. A literature review indicates that the existing methods have provided many relevant tools for SSS problems. However, the majority could benefit from the further exploration of the interrelationships among the selection criteria for a more in-depth analysis. The DEMATEL method is used to determine the degrees of influence among the criteria and solve the criteria issues, namely, dependence and feedback. However, DEMATEL cannot determine the weights of individual criteria, whereas ANP can. ANP was created on the basis of AHP in order to consider the existence of interdependence among the criteria in the model [43]. ANP breaks down problems into clusters, each of which contains a number of variables or criteria to be evaluated. Notably, traditional ANP techniques assume equal weighting. Both methods can

be added to improve and enhance the solution through the interrelationships among the criteria. Liou et al. [44] established a hybrid model for supplier selection of green supply chains in Taiwanese electronic companies by combining DEMATEL-based ANP and a complex proportional assessment of alternatives with gray relations. Their results prove that the obtained weights in each criterion are more reasonable and consistent with the DEMATEL results. Chen et al. [45] used DEMATEL to build an influence network relation and modified ANP to determine the influential weights. Moreover, VIKOR was used to improve wetland environmental management. Zhou and Xu [17] introduced an integrated evaluation model, including the DEMATEL-ANP-VIKOR, to evaluate sustainable supplier selection. By combining fuzzy Delphi and ANP methods, Kannan [46] developed a tool to identify the critical attributes of the suppliers' sustainable compliance. In a case study of an electronic goods manufacturing company, Awasthi et al. [47] utilized fuzzy ANP to generate the criteria weights and fuzzy VIKOR was used to rank the sustainable global supplier selection.

　　Although the MCDM method is more popular in the field of sustainable supplier selection, it is always used to determine the relative importance weights of each criterion. The conventional MCDM methods do not consider uncertain information in human judgments, which may make the decision-makers to obtain the partial relationships among alternatives [48]. To fill this research gap, the present study proposes a novel MCDM model in order to enhance the decision-making process and implement a more in-depth analysis of the supplier selections among the considered plausible interrelationships criteria in uncertain environments.

### 2.2.2. MCDM methods with IFS for SSS

　　In general, uncertainty in the decision-making process is unavoidable [49]. Decision-makers may have different levels of experience, skill, and manner features. Therefore, when they lack the knowledge or experience of the decision information problem, they cannot determine the precise preferences [50]. In order to solve such challenges in the decision-making processes, the fuzzy set theory proposed by Zadeh (1965) can be used for the linguistic term in decision-making processes [51]. Most of the existing studies integrated the fuzzy concept into the traditional MCDM [8]. Wang et al. [52] proposed fuzzy AHP and considered green data envelopment analysis (DEA) within the sustainable supplier selection framework of the SMEs in the food processing industry. Rashidi and Cullinane [53] studied the sustainable supplier selection by comparing fuzzy DEA with fuzzy TOPSIS methods, in order to ensure the commitment of suppliers to the sustainability concept. A considerable amount of literature has highlighted the application of traditional fuzzy set theory combined with various MCDM techniques. However, less attention is paid to IFS. IFS is a generalization of the fuzzy set concept and is ideal for handling real-world cases compared to the classical fuzzy sets. IFS has efficacy in representing uncertainty and vagueness in membership, non-membership and hesitancy values [54]. To achieve the advantage of IFS methods, some studies are reviewed. In a case study on the automotive industry, Memari et al. [55] utilized an intuitionistic fuzzy TOPSIS method to select the right sustainable supplier. Sen et al. [38] applied three decision-making approaches combined with IFS, which overcame the imprecision of human judgment and encouraged supplier selection in a sustainable supply chain. Krishankumar et al. [54] solved the problem of supplier selection with linguistic preferences by extending the intuitionistic fuzzy set-based preference ranking organization method for enrichment (PROMETHEE). Çalı et al. [56] integrated elimination and choice translating reality (ELECTRE) and VIKOR in an intuitionistic fuzzy environment to cope with uncertain situations and hesitancy in the supplier evaluation process.

　　However, various tools for evaluating sustainable supplier and available literature review expose several methods. The interdependencies among dimensions and criteria under uncertainty are thoroughly analyzed, and strategies are provided for selecting improved alternatives to reach the desired aspiration levels. However, the ranking and selections not considered. To make up for this limitation, this study builds an IF-DANP-mV model to enhance decision making in a fuzzy environment and conduct more in-depth analysis of the interrelationships among criteria and to help enterprises aid

their suppliers in improving sustainability and enhance enterprises' socio-environmental performance, which is key to achieving sustainable development.

## 3. Methodology

This section will present the methodological steps used for solving the SSS problem. Once the basic IFS knowledge is presented, the methodology and the steps of the proposed IF-DANP-mV model will be explained.

### 3.1. Intuitionistic Fuzzy Set

IFS was initially developed by Atanassov in 1986 [57], which is an extension of the conventional fuzzy set theory (FST) [51,58]. IFS is beneficial for solving problems involving uncertainty and vagueness [48,49,59]. Defining A is IFS in a finite set X, IFS A is defined as:

$$A = \{\langle x, \mu_A(x), v_A(x)\rangle | x \in X\} \tag{1}$$

where $\mu_A(x) : X \to [0,1]$ and $v_A(x) : X \to [0,1]$ are the respective membership and non-membership functions of the element $x \in X$ in the following condition:

$$0 \le \mu_A(x) + v_A(x) \le 1 \tag{2}$$

In IFS compared with classical FST, there is another parameter, which is known as an intuitionistic fuzzy index or so-called "hesitation degree" [54]. Assume that $\pi_A(x)$ is the hesitation degree of the element $x \in X$ to subset A. $\pi_A(x)$ can be denoted as:

$$\pi_A(x) = 1 - \mu_A(x) - v_A(x) \tag{3}$$

It is clear that for every $x \in X$;

$$0 \le \pi_A(x) \le 1 \tag{4}$$

When the value of $\pi_A(x)$ is small, information concerning x is more confident. When the value of $\pi_A(x)$ is high, information regarding x is much more uncertain; while $\mu_A(x) = 1 - v_A(x)$ for every element of the universe, the multiplication operator for IFS is given:

$$A \otimes B = \{\mu_A(x).\mu_B(x), v_A(x) + v_B(x) - v_A(x).v_B(x) | x \in X\} \tag{5}$$

### 3.2. The Basic of DANP and VIKOR Methods

MCDM is a method that simultaneously considers multiple criteria and aids in decision making by estimating the best case for each criterion after sorting limited available cases according to different characteristics or criteria [45,60]. The Science and Human Affairs Program of the Battelle Memorial Institute of Geneva created and improved the DEMATEL method from 1972 to 1976. This method is used to describe the links among complex criteria in terms of causal relationships in the design and analysis of structural models [61]. As a part of the process, a number of complex and interdependent issues must also be solved in relation to the criteria. The ANP method was initially developed in order to avoid the hierarchical constraints in the AHP method [43]. In the current study, ANP combined with DEMATEL was used to calculate the relative weight of criteria for SSS problems [17]. In order to calculate the relative weights of the criteria, the levels of interdependences shown by the criteria are treated as reciprocal values in the case where only conventional ANP was used. However, reciprocal values do not exist for the levels of interdependences of the criteria according to the DEMATEL method. This notion is close to the real system. Finally, Opricovic [62] proposed and developed VIKOR as a tool for ranking alternatives by using the concept of compromise to evaluate the standard of different projects and then it is possible to use the MCDM model along with VIKOR. This technique can arrange

the results sequentially because it is based on the basic concept of positive-ideal (or aspired level) and negative-ideal (or worst level) solutions [4,45].

### 3.3. Building an IF-DANP-mV Model for Sustainable Supplier Selection

This section explains the developed methodology based on combination of DANP and modified VIKOR methods under an intuitionistic fuzzy environment. The methodology mainly consists of 13 steps which are explained in the detail in this section. Figure 1 shows the purpose model of the sustainable supplier selection/evaluation. The detailed methodology of IF-DANP-mV model is presented in the following section:

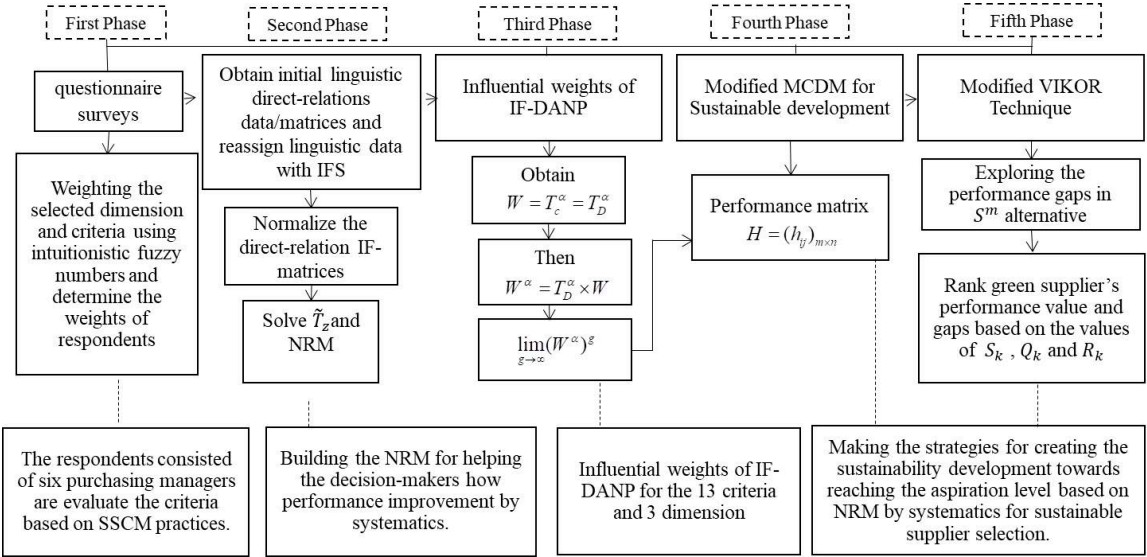

**Figure 1.** Framework for sustainable supplier selection.

**Table 2.** Linguistic terms for ranking the importance of criteria weights.

| Linguistic Terms | Abbreviation | IFS |
|---|---|---|
| No Influence | N | (0.00, 1.00, 0.00) |
| Low Influence | L | (0.35, 0.60, 0.05) |
| Medium Influence | M | (0.50, 0.45, 0.05) |
| High Influence | H | (0.75, 0.20, 0.05) |
| Very High Influence | VH | (0.90, 0.10, 0.00) |

Step 1: Design the IFS linguistic variables

In this step, respondents indicate the linguistic terms as shown in Table 2 to consider the importance degree of each decision-maker. Assume $A_k = [\mu_k, v_k, \pi_k]$ is defined as an intuitionistic fuzzy number for weighting the $k$th decision-maker, calculated as follows:

$$\lambda_k = \frac{\left(\mu_k + \pi_k\left(\frac{\mu_k}{\mu_k + v_k}\right)\right)}{\sum_{k=1}^{l}\left(\mu_k + \pi_k\left(\frac{\mu_k}{\mu_k + v_k}\right)\right)} \quad and, \quad \sum_{k=1}^{l} \lambda_k = 1 \tag{6}$$

Step 2: Create the aggregated intuitionistic fuzzy decision matrix according to the DM opinions.

The process indicated $A^k = (a_{ij}^k)_{m \times n}$ is an intuitionistic fuzzy decision matrix for each DM, and $\lambda = \{\lambda_1, \lambda_2, \lambda_3, \ldots, \lambda_l\}$ is the weight of each DM, where $\sum_{k=1}^{l} \lambda_k = 1$, $\lambda_k \in [0, 1]$. For SSS procedures, all individual DM opinions need to aggregate into a collective form. For this reason, we proposed an intuitionistic fuzzy weighted averaging (IFWA) operator [50] to aggregate the levels of importance

for each criterion. Let $A_{ij}^{(k)} = \left\{ \mu_{ij}^{(k)}, v_{ij}^{(k)}, \pi_{ij}^{(k)} \right\}$ be an IFS that is given by the $k$th DM with the criteria $a_{ij}$. The aggregation process is done by using Equation (7) with the IFWA operator. The aggregated intuitionistic fuzzy relation is formulated as follows:

$$
\begin{aligned}
a_{ij} &= IFWA_\lambda \left( a_{ij}^{(1)}, a_{ij}^{(2)}, \ldots, a_{ij}^{(l)} \right) \\
&= \lambda_1 a_{ij}^{(1)} \oplus \lambda_2 a_{ij}^{(2)} \oplus \lambda_3 a_{ij}^{(3)}, \ldots \oplus \lambda_l a_{ij}^{(l)} \\
&= \left[ 1 - \prod_{k-1}^{l} \left( 1 - \mu_{ij}^{(k)} \right)^{\lambda_k}, \prod_{k-1}^{l} \left( v_{ij}^{(k)} \right)^{\lambda_k}, \prod_{k-1}^{l} \left( 1 - \mu_{ij}^{(k)} \right)^{\lambda_k}, - \prod_{k-1}^{l} \left( v_{ij}^{(k)} \right)^{\lambda_k} \right]
\end{aligned}
\tag{7}
$$

where $a_{ij} = \left( \mu_{Ai}, (x_j), v_{Ai}(x_j), \pi_{Ai}(x_j) \right) (i = 1, 2, \ldots, m; j = 1, 2, \ldots, n)$.

Step 3: Calculate the matrix for the initial influence.

This step is performed to obtain the initial influence matrix $\tilde{A}_z^{(k)}$, which is the calculation of the direct influence exerted by each dimension/criterion $i$ on each dimension/criterion $j$ (shown by degree, i.e., membership, non-membership and hesitancy values). In this set, all principal diagonal elements are equal to zero, and $X$ pertains to the initial effect an element exerts and receives from another element. A contextual relationship among the elements of a system is illustrated using a map. The numeral represents the strength of influence (degree of effect), $z = \left( \mu_{ij}, v_{ij}, \pi_{ij} \right)$.

$$
\tilde{A}_z^{(k)} = \begin{bmatrix}
0 & \tilde{a}_{z12}^{(k)} & \tilde{a}_{z13}^{(k)} & \cdots & \tilde{a}_{z1n}^{(k)} \\
\tilde{a}_{z21}^{(k)} & 0 & \tilde{a}_{z23}^{(k)} & \cdots & \tilde{a}_{z2n}^{(k)} \\
\tilde{a}_{z31}^{(k)} & \tilde{a}_{z32}^{(k)} & 0 & \cdots & \tilde{a}_{z3n}^{(k)} \\
\vdots & \vdots & \vdots & \ddots & \vdots \\
\tilde{a}_{zn1}^{(k)} & \tilde{a}_{zn2}^{(k)} & \tilde{a}_{zn3}^{(k)} & \cdots & 0
\end{bmatrix}
\tag{8}
$$

Step 4: Normalize the direct-relation IF-matrices of membership, non-membership and hesitancy values $\tilde{X}_z^{(k)}$, denoted below:

$$
\tilde{X}_z^{(k)} = l \times \tilde{A}_z^{(k)}
\tag{9}
$$

where

$$
l = \min \left[ \frac{1}{\max \sum_{j=1}^{n} \left| \tilde{A}_z^{(k)} ij \right|}, \frac{1}{\max \sum_{i=1}^{n} \left| \tilde{A}_z^{(k)} ij \right|} \right] i, j \in (1, 2, 3, \ldots, n)
\tag{10}
$$

Step 5: Obtain the total-influence matrix $\tilde{T}_z$.

Equation (11) can be used to obtain the total-influence IF-matrix, where $I$ denote the identity matrix as follows:

$$
\tilde{T}_z = \tilde{X} + \tilde{X}_z^2 + \cdots + \tilde{X}_z^h = \tilde{X}_z \left( I - \tilde{X}_z \right)^{-1}
\tag{11}
$$

When $\lim_{h \to \infty} \tilde{X}_z^i = [0]_{n \times n}$

$$
\tilde{T}_z = \begin{bmatrix}
\tilde{t}_{z11} & \tilde{t}_{z12} & \tilde{t}_{z13} & \cdots & \tilde{t}_{z1n} \\
\tilde{t}_{z21} & \tilde{t}_{z22} & \tilde{t}_{z23} & \cdots & \tilde{t}_{z2n} \\
\tilde{t}_{z31} & \tilde{t}_{z32} & \tilde{t}_{z33} & \cdots & \tilde{t}_{z3n} \\
\vdots & \vdots & \vdots & \ddots & \vdots \\
\tilde{t}_{zn1} & \tilde{t}_{zn2} & \tilde{t}_{zn3} & \cdots & \tilde{t}_{znn}
\end{bmatrix}
\tag{12}
$$

If we define the sum of the rows and columns separately and express them as vectors and $s$, respectively, within the total-influence matrix $\tilde{T}_z$, then Equations (13) and (14) are presented as follows:

$$r = [r_i]_{n\times 1} = \left[\sum_{j=1}^{n} \tilde{t}_{ij}\right]_{n\times 1} \tag{13}$$

$$s = [s_j]_{n\times 1} = \left[\sum_{j=1}^{n} \tilde{t}_{ij}\right]_{1\times n} \tag{14}$$

where the superscript expressions denote transposition.

In addition, $r_i$ shows the sum of the direct and indirect effects of factor $i$ on other factors/criteria if it denotes the sum of the $i$th row in matrix $\tilde{T}_z$. Conversely, $s_i$ represents the sum of the direct and indirect effects that factor $j$ receives from other factors if it denotes the sum of the $j$th column of matrix $\tilde{T}_z$. Furthermore, $(r_i + s_j)$ provides an index of the strength of influences that are given and received when $i = j$ (i.e., sum of row and column aggregates). Specifically, $(r_i + s_j)$ refers to the degree of the role played by factor $i$ in the given problem. In addition, the difference $(r_i - s_j)$ shows the net effect of factor's contributions to the problem. Factor $i$ affects other factors if $(r_i - s_j)$ is positive, whereas it is influenced by other factors if $(r_i - s_j)$ is negative.

Step 6: Generate the unweighted supermatrix.

The total relation matrix $(\tilde{T}_z)$ under the DEMATEL method is used to calculate the relative weight of the criteria and thus avoid the shortcomings identified in the ANP. The DEMATEL method will not solely be used to calculate the level of impacts among groups of criteria. That is, the normalized total-influence matrix will be incorporated into an un-weighted supermatrix $W$ in the ANP, in order to calculate the level of interdependences of different criteria. The supermatrix is a total-influenced matrix that is generated using DEMATEL and uses the sum of each column for normalization. This matrix is denoted as $T_C = [t_{ij}]_{n\times n}$ and $T_C = \left[t_{ij}^{D}\right]_{m\times m}$, which are derived from the criteria and dimensions or clusters, respectively. To normalize supermatrix $T_C$, the weightings from ANP and influence matrix $T_D$ can be applied to the dimensions or clusters. The sum for each column can then be derived for normalization as shown below:

$$T_c = \begin{array}{c} \\ \\ D_1 \\ \\ \vdots \\ D_i \\ \vdots \\ D_n \\ \\ \end{array} \begin{array}{c} c_{11} \\ c_{12} \\ \vdots \\ c_{1m_1} \\ \vdots \\ c_{i1} \\ c_{i2} \\ \vdots \\ c_{im_i} \\ \vdots \\ c_{n1} \\ c_{n2} \\ \vdots \\ c_{nm_n} \end{array} \begin{array}{c} \overset{\begin{array}{ccccccc} D_1 & \cdots & D_j & \cdots & D_n \\ c_{11} \cdots c_{1m_1} & \cdots & c_{ij} & \cdots & c_{jm_j} & \cdots & c_{n1} \cdots c_{nm_n} \end{array}}{\left[\begin{array}{ccccc} T_c^{11} & \cdots & T_c^{1j} & \cdots & T_c^{1n} \\ \vdots & \vdots & \vdots & \vdots & \vdots \\ T_c^{i1} & \cdots & T_c^{ij} & \cdots & T_c^{in} \\ \vdots & \vdots & \vdots & \vdots & \vdots \\ T_c^{nl} & \cdots & T_c^{nj} & \cdots & T_c^{nn} \end{array}\right]} \end{array} \tag{15}$$

The new matrix $T_C^\alpha$ is obtained following the normalization of the total-influence matrix $T_C$ using the dimensions.

$$
T_c^\alpha = \begin{array}{c} \\ \\ \\ \\ \\ D_1 \\ \\ \\ T_c^\alpha = \ D_i \\ \\ \\ D_n \\ \\ \\ \\ \\ \end{array}
\begin{array}{c}
c_{11} \\ c_{12} \\ \vdots \\ c_{1m_1} \\ \vdots \\ c_{i1} \\ c_{i2} \\ \vdots \\ c_{im_i} \\ \vdots \\ c_{n1} \\ c_{n2} \\ \vdots \\ c_{nm_n}
\end{array}
\begin{array}{c}
\begin{array}{ccccccccccc}
& & & D_1 & \cdots & D_j & \cdots & D_n & & \\
c_{11} & \cdots & c_{1m_1} & \cdots & c_{ij} & \cdots & c_{jm_j} & \cdots & c_{n1} & \cdots & c_{nm_n}
\end{array} \\
\left[\begin{array}{ccccc}
T_c^{\alpha 11} & \cdots & T_c^{\alpha 1j} & \cdots & T_c^{\alpha 1n} \\
\vdots & \vdots & \vdots & \vdots & \vdots \\
T_c^{\alpha i1} & \cdots & T_c^{\alpha ij} & \cdots & T_c^{\alpha in} \\
\vdots & \vdots & \vdots & \vdots & \vdots \\
T_c^{\alpha nl} & \cdots & T_c^{\alpha nj} & \cdots & T_c^{\alpha nn}
\end{array}\right]
\end{array}
\tag{16}
$$

Moreover, Equations (17) and (18) explain normalization $T_C^{\alpha 11}$, and other $T_C^{\alpha nm}$ values are denoted as previously shown.

$$
d_{Ci}^{11} = \sum_{j=1}^{m1} t_{cij}^{11}, i, j = 1, 2, \ldots, m_1
\tag{17}
$$

$$
T_D^{\alpha 11} = \left[\begin{array}{ccccc}
t_{c11}^{11}/d_{c1}^{11} & \cdots & t_{c1j}^{11}/d_{c1}^{11} & \cdots & t_{c1m1}^{11}/d_{c1}^{11} \\
\vdots & \vdots & \vdots & \cdots & \vdots \\
t_{ci1}^{11}/d_{ci}^{11} & \cdots & t_{cij}^{11}/d_{ci}^{11} & \cdots & t_{cim1}^{11}/d_{ci}^{11} \\
\vdots & \vdots & \vdots & \ddots & \vdots \\
t_{cm1}^{11}/d_{cm1}^{11} & \cdots & t_{cm1j}^{11}/d_{cm1}^{11} & \cdots & t_{cm1m1}^{11}/d_{cm1}^{11}
\end{array}\right] = \left[\begin{array}{ccccc}
T_{C11}^{11} & \cdots & T_{C1j}^{11} & \cdots & T_{C1m1}^{11} \\
\vdots & \vdots & \vdots & \cdots & \vdots \\
T_{Ci1}^{11} & \cdots & T_{Cij}^{11} & \cdots & T_{Cim1}^{11} \\
\vdots & \vdots & \vdots & \ddots & \vdots \\
T_{Cm1i}^{11} & \cdots & T_{Cm1j}^{11} & \cdots & T_{Cm1m1}^{11}
\end{array}\right]
\tag{18}
$$

The total-influence matrix is enabled to match and fill into the interdependence clusters. Equation (19) supports this process as an unweighted supermatrix, whereby dimensions or clusters transpose the normalized influence matrix $T_C^\alpha$ (i.e., $W = (T_C^\alpha)'$)

$$
W = \left(T_C^\alpha\right)' = \begin{array}{c} \\ \\ \\ \\ \\ D_1 \\ \\ \\ D_i \\ \\ \\ D_n \\ \\ \\ \\ \\ \end{array}
\begin{array}{c}
c_{11} \\ c_{12} \\ \vdots \\ c_{1m_1} \\ \vdots \\ c_{i1} \\ c_{i2} \\ \vdots \\ c_{im_i} \\ \vdots \\ c_{n1} \\ c_{n2} \\ \vdots \\ c_{nm_n}
\end{array}
\begin{array}{c}
\begin{array}{ccccccccccc}
& & & D_1 & \cdots & D_j & \cdots & D_n & & \\
c_{11} & \cdots & c_{1m_1} & \cdots & c_{ij} & \cdots & c_{jm_j} & \cdots & c_{n1} & \cdots & c_{nm_n}
\end{array} \\
\left[\begin{array}{ccccc}
W^{11} & \cdots & W^{i1} & \cdots & W^{n1} \\
\vdots & \vdots & \vdots & \vdots & \vdots \\
W^{1j} & \cdots & W^{ij} & \cdots & W^{nj} \\
\vdots & \vdots & \vdots & \vdots & \vdots \\
W^{1n} & \cdots & W^{in} & \cdots & W^{nn}
\end{array}\right]
\end{array}
\tag{19}
$$

Equation (20) implies that the matrix between the clusters or criteria is independent in cases where the matrix $W^{11}$ is shown as a blank or zero and that interdependence is lacking. Therefore, the remainder of the abovementioned $W^{nn}$ values is obtained as

$$
W^{11} = \begin{array}{c} \\ c_{11} \\ \vdots \\ c_{1j} \\ \vdots \\ c_{1m_1} \end{array}
\begin{array}{c} c_{11} \quad \cdots \quad c_{i1} \quad \cdots \quad c_{1m_1} \\
\left[ \begin{array}{ccccc}
t_{c11}^{\alpha 11} & \cdots & t_{ci1}^{\alpha 11} & \cdots & t_{cm_1 1}^{\alpha 11} \\
\vdots & \vdots & \vdots & \cdots & \vdots \\
t_c & \cdots & t_{cij}^{\alpha 11} & \cdots & t_{cm_1 j}^{\alpha 11} \\
\vdots & \vdots & \vdots & \ddots & \vdots \\
t_{c1m_1}^{\alpha 11} & \cdots & t_{cim_1}^{\alpha 11} & \cdots & t_{cm_1 m_1}^{\alpha 11}
\end{array} \right]
\end{array}
\tag{20}
$$

Step 7: To obtain the weighted supermatrix, each column is summarized for normalization using the Equation (21).

$$
T_D = \begin{bmatrix}
t_D^{11} & \cdots & t_D^{1i} & \cdots & t_D^{1n} \\
\vdots & \vdots & \vdots & \cdots & \vdots \\
t_D^{i1} & \cdots & t_D^{ij} & \cdots & t_D^{in} \\
\vdots & \vdots & \vdots & \ddots & \vdots \\
t_D^{n1} & \cdots & t_D^{nj} & \cdots & t_D^{nn}
\end{bmatrix}
\tag{21}
$$

Equation (22) illustrates the normalization of the total-influence matrix $T_D$ to obtain a new matrix $T_D^{\alpha}$ given by

$$
T_D^{\alpha} = \begin{bmatrix}
t_D^{11}/d_1 & \cdots & t_D^{1j}/d_1 & \cdots & t_D^{1n}/d_1 \\
\vdots & \vdots & \vdots & \cdots & \vdots \\
t_D^{i1}/d_1 & \cdots & t_D^{ij}/d_1 & \cdots & t_D^{in}/d_1 \\
\vdots & \vdots & \vdots & \ddots & \vdots \\
t_D^{n1}/d_1 & \cdots & t_D^{nj}/d_1 & \cdots & t_D^{nn}/d_1
\end{bmatrix}
= \begin{bmatrix}
T_{C11}^{\alpha 11} & \cdots & T_{C1j}^{\alpha 11} & \cdots & T_{C1m1}^{\alpha 11} \\
\vdots & \vdots & \vdots & \cdots & \vdots \\
T_{Ci1}^{\alpha 11} & \cdots & T_{Cij}^{\alpha 11} & \cdots & T_{Cim1}^{\alpha 11} \\
\vdots & \vdots & \vdots & \ddots & \vdots \\
T_{Cm1i}^{\alpha 11} & \cdots & T_{Cm1j}^{\alpha 11} & \cdots & T_{Cm1m1}^{\alpha 11}
\end{bmatrix}
\tag{22}
$$

where, $t_D^{\alpha ij} = t_D^{ij}/d_i$.

To obtain the weighted supermatrix, let the normalized total-influence matrix $T_D^{\alpha}$ fill into the unweighted supermatrix as follows:

$$
W^{\alpha} = T_D^{\alpha} \times W = \begin{bmatrix}
t_D^{\alpha 11}/W^{11} & \cdots & t_D^{\alpha i1}/W^{i1} & \cdots & t_D^{\alpha n1}/W^{n1} \\
\vdots & \vdots & \vdots & \cdots & \vdots \\
t_D^{\alpha 1j}/W^{1j} & \cdots & t_D^{\alpha ij}/W^{ij} & \cdots & t_D^{\alpha nj}/W^{nj} \\
\vdots & \vdots & \vdots & \ddots & \vdots \\
t_D^{\alpha 1n}/W^{1n} & \cdots & t_D^{\alpha in}/W^{in} & \cdots & t_D^{\alpha nn}/W^{nn}
\end{bmatrix}
\tag{23}
$$

Step 8: Limit the weighted supermatrix by raising it to a sufficiently large power $k$.

Continue this step until the supermatrix has converged and converted into a long-term stable supermatrix to obtain global priority vectors. Furthermore, IF-DANP influential weights, such as $\lim_{g \to \infty} (W^{\alpha})^g$ can be obtained. In other words, the limit supermatrix $W^{\alpha}$ with power $g$ (with $g$ representing any number for power) can be used to identify the influential weights of ANP. In summary, the abovementioned steps can be used to derive a stable limiting supermatrix and calculate the overall weights. Therefore, IF-DANP methods can address interdependence and feedback.

Step 9: Using VIKOR for ranking sustainable supplier performance.

The modified VIKOR method can be used for performance improvement multiple alternatives that are influenced by the interaction of various criteria. Equation (24) describes the VIKOR method-based modification for adjusting the IF-DANP matrix. This method assumes that the alternatives are expressed as $S^1$, $S^2$,..., $S^k$,..., $S^m$. In addition, $f_{kj}$ denotes the performance scores of alternative $S^k$ and the $j$th criteria; $w_j$ refers to the influential weight (relative importance) of the $j$th criterion, where $j = 1, 2, \ldots n$ and $n$ represents the number of criteria. The following form of the $L_{p-}$ metric was used to initiate the development of the VIKOR method:

$$L_k^p = \left\{ \sum_{j=1}^n \left[ \frac{w_j \left( \left| f_j^* - f_{kj} \right| \right)}{\left( \left| f_j^* - f_j^- \right| \right)} \right] \right\}^{1/p} \tag{24}$$

where $1 \le p \le \infty; k = 1, 2, \ldots, m$ and the influential weight $W_j$ is derived from the IF-DANP. VIKOR uses $L_k^{p=1}$ (as $S_k$) and $L_k^{p=\infty}$ (as $Q_k$) to formulate the ranking and gap measure, which are respectively given by

$$S_k = L_k^{p=1} = \sum_{j=1}^n \left[ \frac{w_j \left( \left| f_j^* - f_{kj} \right| \right)}{\left( \left| f_j^* - f_j^- \right| \right)} \right] \tag{25}$$

$$Q_k = L_k^{p=\infty} = \max_j \left\{ \frac{\left( \left| f_j^* - f_{kj} \right| \right)}{\left( \left| f_j^* - f_j^- \right| \right)} \, j = 1, 2, \ldots, n \right\} \tag{26}$$

The compromised solution $\min_k L_k^p$ will be selected because it identifies the synthesized gap to be minimized, such that its value will be the closest to the aspired level. In addition, if $p$ is small, then group utility is emphasized (e.g., as $p = 1$). On the contrary, the individual maximal regrets/gaps gain rising importance in prior improvements in each dimension/criteria if $p$ tends to become infinite. Consequently, $\min_k S_k$ stresses the maximum group utility. In comparison, in order to show improvement in priority, $\min_k Q_k$ accentuates the selection of the minimum from maximum individual regrets/gaps.

Step 10: Obtain an aspired or tolerable level.

For all criterion functions, $j = 1, 2, \ldots, n$ we calculated the best ($f_j^*$) (aspired) and worst ($f_j^-$) (tolerable) values. Suppose that the $j$th function denotes benefits, that is $f_j^* = \max_k f_{kj}$ and $f_j^- = \max_k f_{kj}$. Alternatively, these values can be set by decision makers (i.e., $f_j^*$ and $f_j^-$ denote the aspired and tolerable levels, respectively). Furthermore, Equation (27) can be used to convert an original-rating into a normalized weight-rating matrix as follows:

$$r_{kj} = \frac{\left( |f_j^* - f_{kj}| \right)}{\left( |f_j^* - f_j^-| \right)} \tag{27}$$

Step 11: Calculate the mean of group utility and maximal regret.

The values can be computed as follows:

$$S_k = \sum_{j=1}^n w_j r_{kj} \tag{28}$$

As a significant component of the synthesized gap for all criteria, selecting suitable green suppliers is therefore necessary. In addition,

$$Q_k = \max_j \left\{ r_{kj} \middle| j = 1, 2, \ldots, n \right\} \tag{29}$$

Pertain to the maximal gap in the $k$ criterion for priority improvement.

Step 12: Calculate the index value.

The value can be counted by following the formula:

$$R_k = v \frac{(S_k - S^*)}{(S^- - S^*)} + (1 - v) \frac{(Q_k - Q^*)}{(Q^- - Q^*)} \tag{30}$$

where, $k = 1, 2, \ldots, m$, $S^* = \min_i S_i$ "or" $S^* = 0$ (worst-case scenario). Additionally, $Q^* = \min_i Q_i$ or setting $Q^* = 0$ and $Q^- = \max_i Q_i$ or setting $Q^- = 1$, and $v$ is presented as the weight of the strategy of the maximum group utility. Conversely, $1 - v$ is the weight of individual regret. Therefore, when $S^* = 0$, $S^- = 1$, $Q^* = 0$ and $Q^- = 1$, we can re-write $R_k = vS_k + (1-v)Q_k$.

Step 13: Rank or improve the alternatives for a compromise solution.

Here, the alternatives are arranged in descending order based on the values of $S_k$, $Q_k$ and $R_k$. Furthermore, alternatives $S^1$, $S^2$..., $S^M$ are proposed as a compromised solution. The compromise-ranking method, namely, VIKOR is applied to determine the compromised solution. The proposed solution should be adaptable for decision-makers so that it can offer the maximum group utility of the majority (shown by min$S$) and maximal regret of minimum individuals of the opponent (shown by min $Q$).

## 4. Case Study

### 4.1. Problem Description

Palm oil is an important feedstock for biofuel and food. Hence, the global demand for palm oil has increased, with more than doubled production outputs since 2005. Thailand is the world's third-largest palm oil-producing country in the world, accounting for approximately 3% of total global production [25]. Furthermore, the Thai government has promoted palm oil production for many years due to the rising demand for palm oil for food, cosmetics, and especially biodiesel as well as, developed the palm oil products industry toward SSCM [63]. Therefore, the industry is currently characterized by an extremely poor environmental reputation due to the problems associated with palm oil production. Meanwhile, there is still a lack in the consideration of social performance, for example, a lack of evaluation of the employee benefits and the increase in turnover rate. Enterprises rely on suppliers for raw materials, and as such, supplier performance affects the business. In certain cases, enterprises have exerted pressure on suppliers to enhance environmental performance based on SSCM practices, while also strengthening the cooperation between suppliers and customers, in order to improve sustainability in the whole supply chain [64]. Our study demonstrates how an enterprise, Thailand Palm Oil Products Company, utilizes our model to accurately select sustainable suppliers based on SSCM practices. We assign "ABC" as the company name and $S^1$, $S^2$, and $S^3$ as three of its potential suppliers.

### 4.2. Proposed Criteria for SSS-Based SSCM Practices

To identify the sustainable supplier selection criteria, data were preliminary obtained from a comprehensive literature review including papers published in recent years (on the "topic, abstract and keywords" in regards to "sustainable supplier and SSCM practices"), as well as in-depth consultations with experts in the palm oil products industry; based on SSCM practices, a comprehensive measure

for SSS in the Thailand palm oil industry was formulated. Then, we assessed and weighed the importance of each criterion. Finally, the experts confirmed the validity of the 3 dimensions such as economic, environmental and social and a total of 13 criteria were derived from these dimensions (see in Table 1). In order to maintain the long-term benefits of the SSCM through cooperation with partners, this proposed measure seeks to truly reflect the actual efficacy and with the potential of the alternative suppliers. Figure 2 shows the hierarchical representation of selecting the best sustainable supplier.

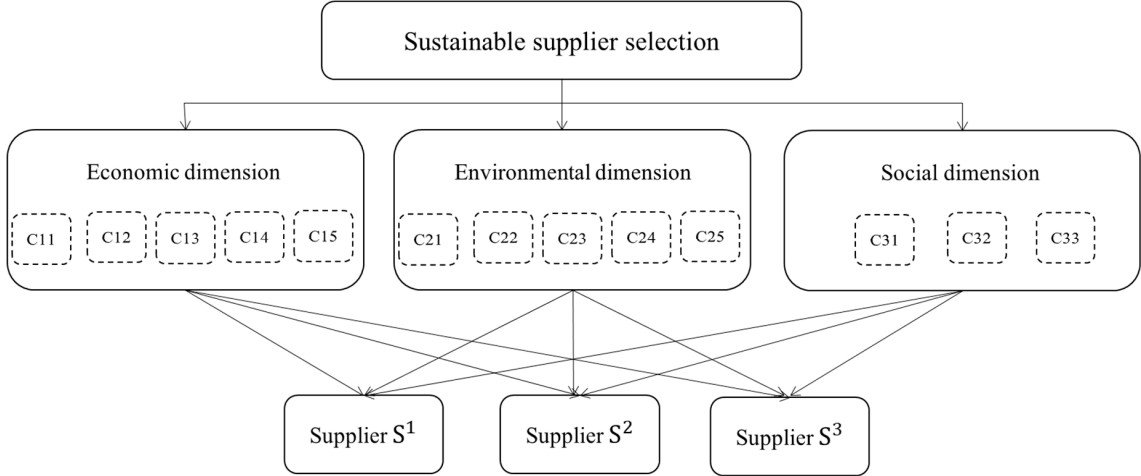

**Figure 2.** The hierarchy of sustainable supplier selection.

### 4.3. Using the IF-DANP-mV Model SSS in a Case Study

This study conducted semi-structured interviews with the respondents that consisted of six purchasing managers with experience in purchasing in the Thai palm oil sector for over ten years to explain and score the direct and indirect influences between the criteria listed. The scale of pairwise comparison of the influential relationship uses the expert's linguistic evaluation interpreting the six experts' responding to linguistic terms in Table 2 and using Equation (6). Then, by using the IFWA operator and Equation (7), experts' judgment aggregated into a collective form can be found in Table 3. Then, matrix $\tilde{A}_z^{(k)}$ is an average initial direct-relation matrix, which is obtained by pairwise comparisons in terms of influences and directions between criteria. Next, Equations (8) and (9) are used to calculate the normalized direct-influence matrix $\tilde{X}_z^{(k)}$, as shown in Table 4. Then, Equations (13) and (14) are used to derive the total influence, $T_C$ and $T_D$, as shown in Table 5. Finally, we obtained the network relation map (NRM) constructed by $r$ and $s$ in the total direct influential matrix, $T_C$ and $T_D$, respectively. These are shown in Figures 3 and 4.

**Table 3.** One expert's linguistic among criteria.

| Criteria | $C_{11}$ | $C_{12}$ | $C_{13}$ | $C_{14}$ | $C_{15}$ | $C_{21}$ | $C_{22}$ | $C_{23}$ | $C_{24}$ | $C_{25}$ | $C_{31}$ | $C_{32}$ | $C_{33}$ |
|---|---|---|---|---|---|---|---|---|---|---|---|---|---|
| $C_{11}$ | N | M | H | N | VH | H | L | M | VH | VH | H | M | H |
| $C_{12}$ | L | N | VH | N | H | L | H | H | M | L | M | H | VH |
| $C_{13}$ | M | L | N | H | M | H | L | VH | VH | H | M | N | M |
| $C_{14}$ | H | VH | H | N | M | L | M | H | VH | M | L | L | M |
| $C_{15}$ | VH | H | L | M | N | M | VH | L | H | H | VH | M | H |
| $C_{21}$ | M | VH | H | M | H | N | L | H | VH | N | L | H | L |
| $C_{22}$ | H | VH | M | M | H | L | N | H | VH | M | M | H | VH |
| $C_{23}$ | M | L | H | VH | VH | H | M | N | H | H | M | VH | M |
| $C_{24}$ | H | H | L | M | H | VH | VH | M | N | M | H | M | H |
| $C_{25}$ | M | L | H | M | L | H | VH | M | N | N | H | M | M |
| $C_{31}$ | H | VH | VH | M | H | VH | H | M | N | L | N | H | VH |
| $C_{32}$ | L | H | M | H | VH | H | L | N | M | H | VH | N | M |
| $C_{33}$ | H | VH | M | L | L | M | H | M | VH | VH | M | L | N |

**Table 4.** Direct-influence matrix $\tilde{X}_z^{(k)}$ for criteria after normalization.

| Criteria | $C_{11}$ | $C_{12}$ | $C_{13}$ | $C_{14}$ | $C_{15}$ | $C_{21}$ | $C_{22}$ | $C_{23}$ | $C_{24}$ | $C_{25}$ | $C_{31}$ | $C_{32}$ | $C_{33}$ |
|---|---|---|---|---|---|---|---|---|---|---|---|---|---|
| $C_{11}$ | 0 | 0.052 | 0.039 | 0 | 0.054 | 0.021 | 0.063 | 0.022 | 0.082 | 0.043 | 0.012 | 0.033 | 0.055 |
| $C_{12}$ | 0.081 | 0 | 0.090 | 0 | 0.071 | 0.032 | 0.071 | 0.094 | 0.080 | 0.068 | 0.034 | 0.040 | 0.060 |
| $C_{13}$ | 0.072 | 0.043 | 0 | 0.034 | 0.055 | 0.067 | 0.057 | 0.051 | 0.042 | 0.044 | 0.066 | 0 | 0.012 |
| $C_{14}$ | 0.050 | 0.020 | 0.031 | 0 | 0.013 | 0.021 | 0.081 | 0.060 | 0.055 | 0.029 | 0.087 | 0.011 | 0.055 |
| $C_{15}$ | 0.040 | 0.054 | 0.043 | 0.055 | 0 | 0.029 | 0.092 | 0.033 | 0.075 | 0.043 | 0.080 | 0.082 | 0.009 |
| $C_{21}$ | 0.023 | 0.073 | 0.067 | 0.018 | 0.010 | 0 | 0.070 | 0.011 | 0.059 | 0 | 0.023 | 0.078 | 0.090 |
| $C_{22}$ | 0.035 | 0.053 | 0.071 | 0.049 | 0.036 | 0.043 | 0 | 0.044 | 0.011 | 0.064 | 0.020 | 0.092 | 0.019 |
| $C_{23}$ | 0.045 | 0.040 | 0.071 | 0.011 | 0.060 | 0.066 | 0.081 | 0 | 0.040 | 0.036 | 0.044 | 0.014 | 0.043 |
| $C_{24}$ | 0.039 | 0.066 | 0.003 | 0.073 | 0.090 | 0.058 | 0.030 | 0.067 | 0 | 0.070 | 0.051 | 0.035 | 0.012 |
| $C_{25}$ | 0.041 | 0.038 | 0.048 | 0.048 | 0.014 | 0.089 | 0.048 | 0.051 | 0 | 0 | 0.063 | 0.016 | 0.088 |
| $C_{31}$ | 0.033 | 0.081 | 0.090 | 0.090 | 0.035 | 0.094 | 0.061 | 0.022 | 0 | 0.022 | 0 | 0.031 | 0.030 |
| $C_{32}$ | 0.067 | 0.062 | 0.011 | 0.012 | 0.049 | 0.030 | 0.030 | 0 | 0.059 | 0.013 | 0.054 | 0 | 0.019 |
| $C_{33}$ | 0.044 | 0.082 | 0.021 | 0.026 | 0.038 | 0.014 | 0.091 | 0.033 | 0.060 | 0.045 | 0.022 | 0.004 | 0 |

**Table 5.** The sum of influential matrices $T_D$ and $T_C$ for the dimension and criteria.

| | $T_D$ | | | | | $T_C$ | | | |
|---|---|---|---|---|---|---|---|---|---|
| Dimension | $r_i$ | $s_i$ | $r_i + s_i$ | $r_i - s_i$ | Criteria | $r_i$ | $s_i$ | $r_i + s_i$ | $r_i - s_i$ |
| $C_1$ | 1.994 | 1.751 | 3.745 | 0.243 | $C_{11}$ | 0.985 | 1.087 | 2.072 | −0.102 |
| | | | | | $C_{12}$ | 0.053 | 1.245 | 1.298 | −1.192 |
| | | | | | $C_{13}$ | 0.866 | 0.504 | 1.370 | 0.362 |
| | | | | | $C_{14}$ | 1.204 | 1.036 | 2.240 | 0.168 |
| | | | | | $C_{15}$ | 1.056 | 1.098 | 2.154 | −0.042 |
| $C_2$ | 1.908 | 1.984 | 4.892 | −0.076 | $C_{21}$ | 1.300 | 1.041 | 2.341 | 0.259 |
| | | | | | $C_{22}$ | 0.981 | 0.910 | 1.891 | 0.071 |
| | | | | | $C_{23}$ | 0.654 | 0.694 | 1.348 | −0.040 |
| | | | | | $C_{24}$ | 0.851 | 0.866 | 1.717 | −0.015 |
| | | | | | $C_{25}$ | 0.774 | 0.496 | 1.270 | 0.278 |
| $C_3$ | 1.854 | 1.878 | 3.732 | −0.024 | $C_{31}$ | 0.803 | 0.793 | 1.596 | 0.010 |
| | | | | | $C_{32}$ | 0.835 | 0.919 | 1.754 | −0.084 |
| | | | | | $C_{33}$ | 0.699 | 0.861 | 1.560 | −0.162 |

Note: Let $i = j$ be $r_i + s_i$ and $r_i - s_i$.

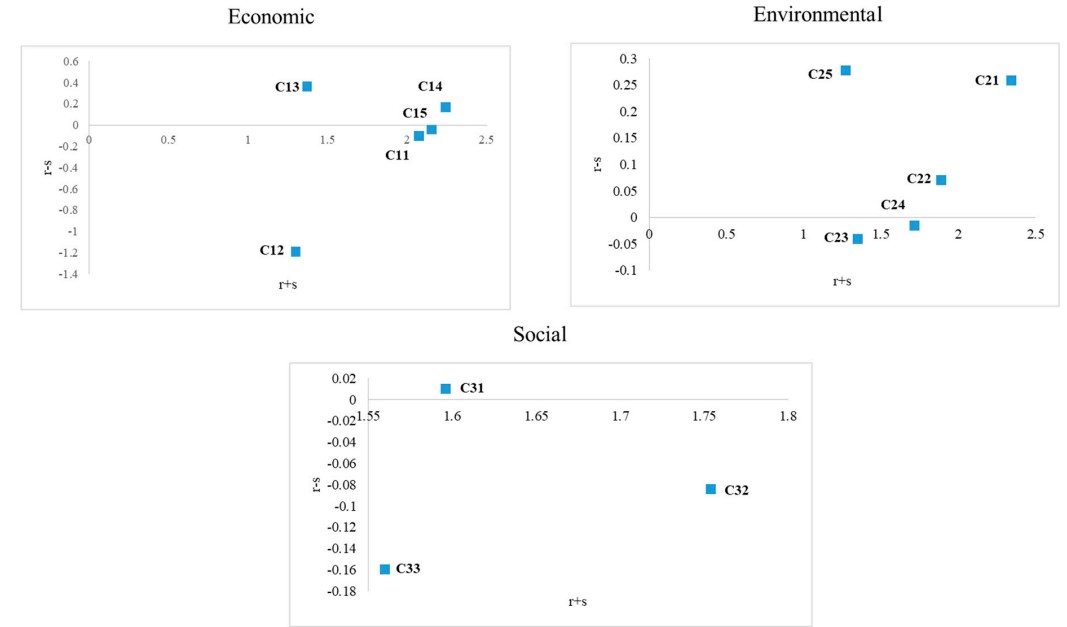

**Figure 3.** Influential NRM of 13 criteria within 3 dimensions.

After confirming the interfering relationship with the criteria, we structured the IF-DANP method by calculating the substantial weight of sustainable supplier criteria. First, an unweighted supermatrix $W$ can be obtained by $\left(T_C^\alpha\right)'$. Then, we achieved the weighted supermatrix $W^\alpha$ using Equation (23) shown in Table 6 in accordance with the extent of the impacts of various criteria. Finally, $\lim_{g\to\infty}\left(W^\alpha\right)^g$ was derived after calculating the limiting power of the weighted supermatrix (see Table 7). The limit for the supermatrix confirms that it converged to form a long-term stable supermatrix—one that is capable of providing the rankings and the local and global weights for the selected criteria.

**Table 6.** Weighting the unweighted supermatrix based on the total-influence normalized matrix $W^\alpha$.

| Criteria | $C_{11}$ | $C_{12}$ | $C_{13}$ | $C_{14}$ | $C_{15}$ | $C_{21}$ | $C_{22}$ | $C_{23}$ | $C_{24}$ | $C_{25}$ | $C_{31}$ | $C_{32}$ | $C_{33}$ |
|---|---|---|---|---|---|---|---|---|---|---|---|---|---|
| $C_{11}$ | 0.071 | 0.062 | 0.082 | 0.066 | 0.08 | 0.062 | 0.052 | 0.063 | 0.056 | 0.052 | 0.049 | 0.054 | 0.068 |
| $C_{12}$ | 0.067 | 0.051 | 0.063 | 0.062 | 0.077 | 0.073 | 0.084 | 0.074 | 0.058 | 0.073 | 0.063 | 0.053 | 0.058 |
| $C_{13}$ | 0.058 | 0.063 | 0.056 | 0.053 | 0.048 | 0.079 | 0.058 | 0.048 | 0.073 | 0.056 | 0.078 | 0.069 | 0.069 |
| $C_{14}$ | 0.082 | 0.074 | 0.082 | 0.074 | 0.052 | 0.071 | 0.081 | 0.084 | 0.054 | 0.072 | 0.059 | 0.078 | 0.048 |
| $C_{15}$ | 0.062 | 0.072 | 0.08 | 0.062 | 0.067 | 0.081 | 0.066 | 0.071 | 0.076 | 0.061 | 0.071 | 0.082 | 0.052 |
| $C_{21}$ | 0.069 | 0.08 | 0.071 | 0.079 | 0.07 | 0.065 | 0.072 | 0.058 | 0.069 | 0.075 | 0.085 | 0.055 | 0.075 |
| $C_{22}$ | 0.086 | 0.081 | 0.081 | 0.061 | 0.08 | 0.085 | 0.055 | 0.065 | 0.061 | 0.088 | 0.058 | 0.085 | 0.075 |
| $C_{23}$ | 0.061 | 0.061 | 0.051 | 0.077 | 0.063 | 0.041 | 0.04 | 0.049 | 0.067 | 0.066 | 0.065 | 0.063 | 0.066 |
| $C_{24}$ | 0.074 | 0.074 | 0.051 | 0.074 | 0.073 | 0.058 | 0.06 | 0.07 | 0.074 | 0.071 | 0.061 | 0.055 | 0.061 |
| $C_{25}$ | 0.053 | 0.065 | 0.058 | 0.055 | 0.055 | 0.058 | 0.078 | 0.074 | 0.065 | 0.068 | 0.07 | 0.068 | 0.068 |
| $C_{31}$ | 0.083 | 0.083 | 0.083 | 0.073 | 0.073 | 0.063 | 0.073 | 0.073 | 0.073 | 0.083 | 0.085 | 0.073 | 0.083 |
| $C_{32}$ | 0.058 | 0.058 | 0.068 | 0.064 | 0.064 | 0.058 | 0.059 | 0.059 | 0.064 | 0.061 | 0.071 | 0.058 | 0.058 |
| $C_{33}$ | 0.056 | 0.046 | 0.046 | 0.046 | 0.058 | 0.066 | 0.072 | 0.072 | 0.056 | 0.046 | 0.056 | 0.067 | 0.069 |

Note: $W^\alpha = T_D^\alpha \times W$.

**Table 7.** Stable matrix of IF-DANP when power $\lim_{g\to\infty}\left(W^\alpha\right)^g$.

| Criteria | $C_{11}$ | $C_{12}$ | $C_{13}$ | $C_{14}$ | $C_{15}$ | $C_{21}$ | $C_{22}$ | $C_{23}$ | $C_{24}$ | $C_{25}$ | $C_{31}$ | $C_{32}$ | $C_{33}$ |
|---|---|---|---|---|---|---|---|---|---|---|---|---|---|
| Weight (IF-DANP) | 0.062 | 0.072 | 0.065 | 0.081 | 0.080 | 0.082 | 0.073 | 0.072 | 0.083 | 0.070 | 0.055 | 0.053 | 0.060 |

Using VIKOR to evaluate total performance, we used $S^1$, $S^2$, and $S^3$ to denote the three suppliers. We assessed the suppliers' performance by gathering insights from six experts from the Thai palm oil sector. A rating scale that ranged from 0 to 4 was used for the evaluation, where 0 represents very low performance, whereas 1 denotes very high performance. The mean scores were taken for each supplier. After applying Equations (28)–(30), the VIKOR technique was applied to determine the indices for ranking, namely, $Q_k$, $R_k$ and $S_k$. We further utilized the technique to explore the gaps in the aspired level with regard to alternative suppliers and as indicated in Table 8.

*4.4. Sensitivity Analysis*

In this section, a sensitivity analysis was conducted to test the strength of the proposed framework. Sensitivity analysis aids to determine whether the variation of the criteria relative weights may or may not change in the final ranking of the alternatives. The results of sensitivity analysis in this study suggest that $S^3$ has the highest rank among all alternatives when $v$ varies from 0.1 to 1.0 (see Table 9). However, after an increase in the $v$ value, we can see that the ranking of alternative using IF-DANP-mV model does not affect the ranking results and it indicates that $S^3$ outperforms $S^2$ and $S^1$, as exhibited in Figure 4. This potential of the IF-DANP-mV method can assist decision-makers more meaningfully and judgmentally to evaluate the sustainable supplier in the supply chain.

**Table 8.** Gaps in performance and aspired level of each sustainable suppliers.

| Dimensions | Criteria | Weight | Gap of Aspired Level | | |
|---|---|---|---|---|---|
| | | | $S^1$ | $S^2$ | $S^3$ |
| $C_1$ | $C_{11}$ | 0.062 | 0.985 | 1.087 | 2.072 |
| | $C_{12}$ | 0.072 | 0.053 | 1.245 | 1.298 |
| | $C_{13}$ | 0.065 | 0.866 | 0.504 | 1.370 |
| | $C_{14}$ | 0.081 | 1.204 | 1.036 | 2.240 |
| | $C_{15}$ | 0.080 | 1.056 | 1.098 | 2.154 |
| $C_2$ | $C_{21}$ | 0.082 | 1.300 | 1.041 | 2.341 |
| | $C_{22}$ | 0.073 | 0.981 | 0.910 | 1.891 |
| | $C_{23}$ | 0.072 | 0.654 | 0.694 | 1.348 |
| | $C_{24}$ | 0.083 | 0.851 | 0.866 | 1.717 |
| | $C_{25}$ | 0.070 | 0.774 | 0.496 | 1.270 |
| $C_3$ | $C_{25}$ | 0.070 | 0.774 | 0.496 | 1.270 |
| | $C_{31}$ | 0.052 | 0.803 | 0.793 | 1.596 |
| | $C_{32}$ | 0.053 | 0.835 | 0.919 | 1.754 |
| | $C_{33}$ | 0.060 | 0.699 | 0.861 | 1.560 |
| Total Gap Performance Scores(Ranking) | | | 2.541(3) | 2.582(2) | 3.095(1) |
| $S_k$ | | | 0.566(3) | 0.493(2) | 0.453(1) |
| $R_k$ | | | 0.050(3) | 0.055(2) | 0.031(1) |

**Table 9.** The sensitivity runs for alternatives when $v$ varies from 0.1 to 1.0.

| Alternatives | $v = 0.1$ | Rank | $v = 0.2$ | Rank | $v = 0.3$ | Rank | $v = 0.4$ | Rank | $v = 0.5$ | Rank |
|---|---|---|---|---|---|---|---|---|---|---|
| $S^1$ | 0.030 | 3 | 0.038 | 3 | 0.040 | 3 | 0.055 | 3 | 0.061 | 3 |
| $S^2$ | 0.024 | 2 | 0.032 | 2 | 0.031 | 2 | 0.050 | 2 | 0.043 | 2 |
| $S^3$ | 0.012 | 1 | 0.011 | 1 | 0.020 | 1 | 0.046 | 1 | 0.033 | 1 |
| **Alternatives** | $v = 0.6$ | **Rank** | $v = 0.7$ | **Rank** | $v = 0.8$ | **Rank** | $v = 0.9$ | **Rank** | $v = 1.0$ | **Rank** |
| $S^1$ | 0.063 | 3 | 0.072 | 3 | 0.105 | 3 | 0.010 | 3 | 0.031 | 3 |
| $S^2$ | 0.045 | 2 | 0.060 | 2 | 0.089 | 2 | 0.084 | 2 | 0.055 | 2 |
| $S^3$ | 0.039 | 1 | 0.040 | 1 | 0.071 | 1 | 0.062 | 1 | 0.050 | 1 |

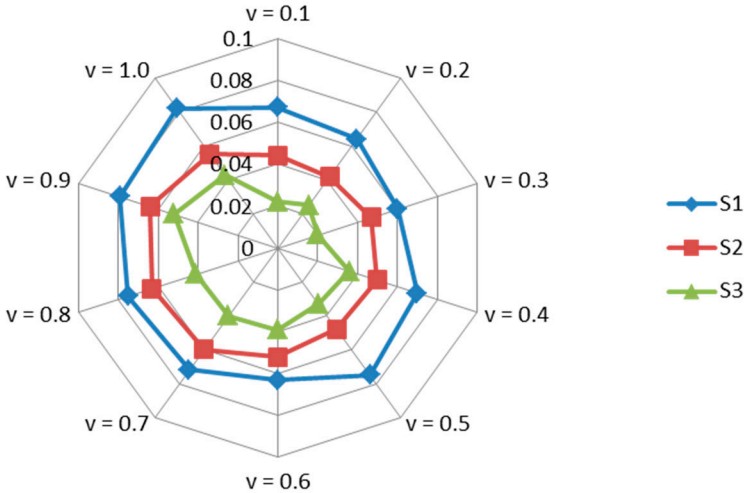

**Figure 4.** Result of sensitivity analysis under the IF-DANP-mV model.

*4.5. Comparisons with Other Existing Methods*

In order to verify the validity of the proposed method, a comparative analysis was conducted with the traditional DANP-VIKOR to prove this example. The results are shown in Table 10. It can

be found that $S^3$ is still the most suitable and sustainable supplier. However, the results obtained by traditional DANP-VIKOR methods demonstrate that the lowest ranking changed from supplier $S^1$ to supplier $S^2$. Detailed comparisons of the results with different methods are explained as follows:

First of all, the comparison is conducted between the crisp numbers and intuitionistic fuzzy numbers. The proposed approach uses the linguistic terms to describe the uncertainty, while traditional DANP-VIKOR uses the crisp number without considering uncertain information in human judgments. For example, if the procurement expert gives the score 5 (very high influence), the score can be transformed into (0.90, 0.10, 0.00). Obviously, in the traditional method, the procurement team can obtain a partial ranking among alternatives, which will interrupt the procurement team to directly identify the efficiency supplier.

Secondly, the proposed model is further supported by the results of sensitivity runs of VIKOR with different weight values. As shown in Figure 5, supplier $S^3$ is also the smallest as $v$ varies from 0.1 to 1.0, but the order with other suppliers changed. Supplier $S^2$ has the orders changed when $v$ varies, which indicates that the percentage of variability is 12% whilst IF-DANP-mV is 8.5%. Therefore, we can see that the traditional DANP-VIKOR method is rather sensitive to the changes in the weight of the evaluation value.

Finally, it is evident that considering uncertainties of evaluations has the potential to solve the uncertainty problem with a different kind of evaluation information compared with the traditional DANP-VIKOR. This merit of the comparative analysis will help the procurement and researcher to analyze and verify the advantages of this method more accurately.

**Table 10.** Comparisons with other methods.

| Methods | Orders of Alternatives |
|---------|------------------------|
| IF-DANP-mV | $S^3 > S^2 > S^1$ |
| DANP-VIKOR | $S^3 > S^1 > S^2$ |

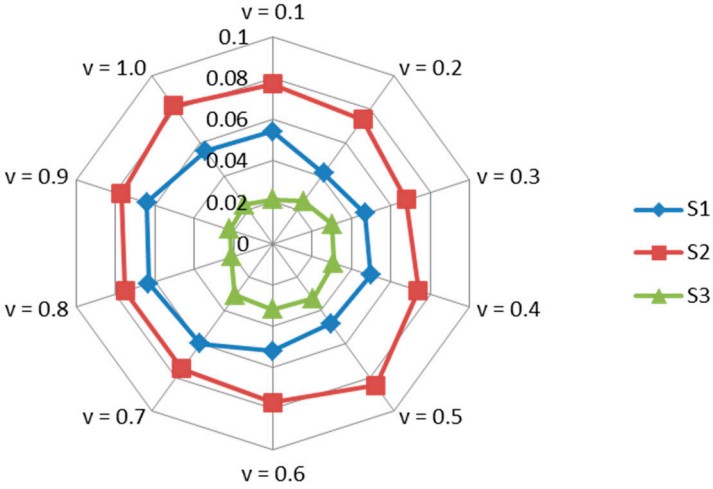

**Figure 5.** Result of sensitivity analysis under the DANP-VIKOR model.

## 5. Results and Discussion

The findings of this study are summarized and discussed as follow:

The result is based on influential relation analysis scores which were determined based on three different sustainability pillars. From the data in Table 5 and Figure 3, we can see the following: Firstly, regarding the economic dimension aspect, based on $(r_i - s_i)$ value we found that supply flexibility $(C_{13})$ and cost reduction activities $(C_{14})$ belong to the causal group, while the effect group consisted of delivery and service of product $(C_{15})$, products' quality improvement $(C_{11})$ and process capability $(C_{12})$. This indicates that $(C_{13})$ has the most significant impact among other criteria, while $(C_{14})$ has

the smallest value. Based on this information, supply flexibility is the main criterion getting more attention among the economic dimensions. For example, changing market demands and differing supplier lead-times are sources of uncertainty that are needed for building supply flexibility. Secondly, regarding the environmental dimension aspect, the result shows that environmental policies ($C_{23}$) and environmental planning ($C_{24}$) are categorized into the effect group and implementation and operation ($C_{25}$); green image ($C_{21}$) and environmentally friendly materials ($C_{22}$) are the cause group. This implies that ($C_{25}$) has the highest score in ($r_i - s_i$) value, meaning that the enterprise wanting to improve the performance of operation products and signaling move cooperation towards sustainable supply chains and sustainable business networks. For a sustainable supplier, it is recommended to push forward implementation and operation in order to expand sales opportunities and extend the long-term relationship with enterprises. Thirdly, regarding the social aspect dimension, in this case, the effect group contains safety and health system ($C_{32}$) and stakeholder relationship ($C_{33}$) while employer rights and welfare ($C_{31}$) is the cause group. It is seen that ($C_{31}$) has a high value of priority compared to the other criteria; we can indicate that the evaluation of the lack of employee benefits and the increasable turnover rate are the critical problems. Therefore, the enterprises have to pay the most attention to employment rights and welfare in order to enhance the social performance in the supply chain. This result was compatible with Zhou and Xu [17]; most experts in the case have recognized that suppliers should pay attention on social responsibility to maintain the sustainability.

Based on the results utilizing IF-DANP, the final weights can be ranked as follows: environmental planning ($C_{24}$) ranks first with the highest weight value (0.083), and safety and health system ($C_{32}$) occupies the last rank with the lowest value (0.053) in all evaluation criteria. Environmental planning ($C_{24}$), green image ($C_{21}$), cost reduction activities ($C_{14}$), delivery and service of the product ($C_{15}$), environmentally friendly materials ($C_{22}$) criteria have been ranked as the top five criteria for SSS based on SSCM practices. Interestingly, from the ranking results, the enterprises are usually more sensitive to environmental planning than social responsibility. Therefore, considering only the score of experience and knowledge, it is difficult to judge the social responsibility, which is in accordance with Zhou and Xu [17].

Furthermore, our findings are applicable to improvements because they can be used to determine gaps in the aspired level of the criteria. Upon examining the findings obtained through the IF-DANP-mV model (see Table 9) the performance results generated values that were arranged as $S^3 > S^2 > S^1$. This finding indicates that the best sustainable supplier in this case study is $S^3$. Moreover, our model also illustrates the means by which alternatives help a company reach its aspiration level for each criterion. Besides, the sensitivity analysis can be significant to evaluate the alternatives for SSS in SSCM practices. In the end, to further verify the validity of our proposed method, we provide realistic evidence for the comparative analysis; the result showed that an integrated DEMATEL-ANP-VIKOR combined with an intuitionistic fuzzy number can provide strategies for selecting improved alternatives to reach the desired aspiration levels apart from the ranking and selections, as also emphasized in Büyüközkan et al. [48].

## 6. Conclusions

Several options to reduce pollution have been introduced in various industries in recent years. In general, solutions for reducing the environmental impacts have been identified, and one of the solutions is the encouragement of sustainable supplier selection based on SSCM practices, in order to improve sustainability in the whole supply chain [65]. Since the requirement of incorporating the sustainable criteria into conventional supplier selection practices is considered and the uncertain decision information affects the decision-maker, which might be a problem of information loss, this study builds a new extension to the MCDM model for supplier selection under a fuzzy environment to enhance decision-making processes and investigates more in-depth analysis of the interrelationships among criteria.

The purpose is presented along with the operational model, which considers suppliers in terms of sustainability to validate a supplier's effectiveness and feasibility. The major contributions and innovations of this paper are summarized as follows:

1. We have the supplier evaluation criteria based on SSCM practice in economic, environmental, and social aspects. In order to investigate suppliers' implementation of SSCM practices, the potential suppliers with sustainability were discovered and selected. The data were obtained by interviewing with procurement experts. The validity of the three dimensions and a total of 13 criteria were confirmed. By constructing the list of evaluation criteria for suppliers and measuring their relative importance, enterprises can better understand the concept of sustainability [66]. Besides, enterprises can be employed for early development of suppliers, which also helps to focus on the target suppliers. Meanwhile, SSCM practices allow suppliers to pay close attention to the area in which they can satisfy the requirements of enterprises.

2. This study extended MCDM under an intuitionistic fuzzy environment, and proposed an IF-DANP-mV model. First, we applied the DEMATEL combined with the intuitionistic fuzzy method to construct a relationship network. Second, the IF-DANP approach was used to calculate the substantial weight of the criteria and overcome the dependence and feedback among the conflicting criteria and uncertain environment. As a result, an intuitionistic fuzzy set is helpful to cope with uncertainty, and it is more flexible to handle precise problems. DANP is a powerful technique which can be used to determine the relative weights of criteria, which is consistent with the results obtained by Govindan et al. [49] and Büyüközkan et al. [48]. Hence, enterprises can effectively enhance decision-making capability. This model has the efficiency to consider uncertainty in human judgments. In order to evaluate the total performance scores and gaps (that is, the smaller, the better) at each aspiration level, the VIKOR concepts were further modified. Our model demonstrates the case study of three supplier candidates, namely, $S^1$, $S^2$ and $S^3$. The VIKOR results indicate the ranking of sustainable supplier in descending order as $S^3 > S^2 > S^1$. A sensitivity analysis was also conducted to test the robustness of the proposed framework. Finally, in order to prove the merit of the proposed approach, a comparison was made with traditional DANP-VIKOR methods, showing that the IF-DANP-mV method performs better than the traditional DANP-VIKOR in dealing with uncertain information.

The benefits of a new extension approach can enhance decision making in a fuzzy environment and a more in-depth analysis of the interrelationships among criteria. By using the proposed approach, suppliers are more accurately ranked when various uncertainties are coped with. Enterprises can analyze the suppliers that have a great difference compared with the other methods. Moreover, the results can help suppliers to discover their weak links and improve their management level. On this basis, a strong relationship can be built between managers and their partners.

3. This work narrows the theoretical gaps identified by Memari et al. [55], who proposed the SSS problem by using a multi-criteria intuitionistic fuzzy TOPSIS model to apply an intuitionistic fuzzy number. However, their results demonstrate that the integration with TOPSIS ignores the interdependencies among criteria which can influence the outcome of the alternative ranking. Therefore, different from the study of Memari et al., the present study extends the MCDM method by applying the DANP to handle the dependencies among decision criteria and modify the VIKOR method to identify the suitable alternative ranking. Our results are more accurate, and the realistic investigation based on a real-world case study is better than using a single method.

With an increasing number of entrepreneurs focusing on sustainable development, especially sustainable supplier management [28], it is a challenge for several industries to consider the selection of sustainable suppliers. This study not only provides an effective model for measuring supplier selection performance, but also plays an audit role for practitioners to evaluate the standardized procedure based on sustainable supply chain management practices.

Although this model can effectively handle uncertainties in decision making, there are still has some limitations. First, this study only identifies one case study in the palm oil product industry, and conclusions may not apply to various industries. Different industries have different products

and procedure characteristics, which can influence SSCM practices. Furthermore, we found that the experts' weights in the decision-making group are not consistent with the reality because their opinions can be effective in the decision-making process. Further study should focus on the understanding of sustainable supplier selection with risk criteria under SSCM performance, in order to help enterprises introduce more effective SSCM practices.

**Author Contributions:** Both authors contributed equally in the writing of this article.

**Funding:** This work was partly supported by National Natural Science Foundation of China (Nos. 71671188).

**Conflicts of Interest:** The authors declare no conflict of interest.

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
