# Peer review of "A New Extension to a Multi-Criteria Decision-Making Model for Sustainable Supplier Selection under an Intuitionistic Fuzzy Environment"

_sustainability, doi:10.3390/su11195413_

Round 1

Reviewer 1 Report

The paper "A New Extension to Multi-Criteria Decision-Making model for Sustainable Supplier Selection under Intuitionistic Fuzzy Environment" has been modified following the reviewers' comments. One can see some improvements, especially in linking the paper to the sustainability concept, but more (mostly) minor improvements are needed.

The paper has not been thoroughly modified but some bits and pieces of text were amended using inserted words. In some other places, the whole paragraphs were added. In general, the Conclusions need to be expanded a bit more in order to explain the results in a more comprehensive way. In addition, English proofreading is still required.

Reviewer 2 Report

The author(s) have submitted a revised manuscript. Some of the elements suggested by the reviewers have been improved but there is, in my opinion, still much work required to ensure this is of a publishable standard.

Overall – while I think the manuscript is interesting and the method (combination of methods) is novel, I am not convinced that this proposed approach is adding value and this has not been clearly demonstrated or explained in the revised manuscript. The manuscript therefore fails the “so what” test for me – it appears, at the end, that the ‘traditional DANP’ approach is just as effective and gives the same results – why shouldn’t a manager skip the complexity of your approach and use this approach?

The introduction has been improved. The motivation, however, remains weak.

While there is a good review of the antecedents to the study (p. 2, lines 47-61), the apparent motivation for the study (p. 2, lines 62-68) has not been clearly explained. At the core, the issue seems to be one of ‘dealing with uncertain and fuzzy decision information’ and the resulting ‘information loss.’ These issues need to be much more clearly identified and explained.
The problem is also apparent in Section 2.2.1 on MCDM for SSS: while there is extensive talking about other MDCM over pages 4-5, the limitations or problems are not highlighted here. I would expect to see a critical discussion of the management of uncertainty, fuzzy decision information, and information loss with a clear explanation of why other methods cannot address these issues.

So far, this section is not convincing. As a result, the rest of the paper is poorly motivated and just seems to be a ‘new combination’ of approaches or techniques that is not solving a real/genuine problem.

This comes back to my previous comment on the comparison with existing methods, covered in Sections 4.4 and 4.5: the results are the same as when the traditional DANP approach is used. Table 10 is redundant – you can simply state the order of alternatives is the same.
p. 17, lines 488-490: if DANP method doesn’t consider the inherent ambiguity in human judgement (this is not one of the motivating factors highlighted in the introduction, by the way, where you focused on uncertain and fuzzy decision information and information loss), but comes up with the same result … what difference does it make? Under what circumstances is your proposed approach going to be more appropriate? If the traditional approach gives the same solution … isn’t the issue of human judgement irrelevant to the problem you are examining?

You talk about the “a different kind of evaluation information without information loss” (p. 17, line 493) but you have not actually demonstrated any difference in how the traditional approach and your proposed approach are different in these ways.

This may be a problem with the particular case you are using to illustrate your method.

Based on the unconvincing results and the poor connection to past research (explaining the problem that this combination of methods will solve) and the lack of difference between these results and the traditional approach in Section 4, I do not believe the manuscript presents sufficient contribution to warrant publication.

It is unclear whether these issues can be addressed with a revision. It may require a substantial new case study and an improved literature review.

Reviewer 3 Report

An interesting paper, but it needs some revisions before taking into account for publication. I provide my main comments below:

in the literature review, I miss reference to some of the key articles in the field of sustainable supply chain management. You need to expand your literature review in order to better frame your research in the existing body of knowledge. Some examples: regarding the conceptualisation of sustainability in supply chain context, one of the key references is Carter, C. R., & Rogers, D. S. (2008). A framework of sustainable supply chain management: moving toward new theory. International journal of physical distribution & logistics management38(5), 360-387. regarding further definition and interpretation of SSCM, see Ahi, P., & Searcy, C. (2013). A comparative literature analysis of definitions for green and sustainable supply chain management. Journal of cleaner production52, 329-341. as well as Seuring, S., & Müller, M. (2008). From a literature review to a conceptual framework for sustainable supply chain management. Journal of cleaner production16(15), 1699-1710. regarding consumer influence in SSCM, see Veit, C., Lambrechts, W., Quintens, L., & Semeijn, J. (2018). The impact of sustainable sourcing on customer perceptions: Association by guilt from scandals in local vs. offshore sourcing countries. Sustainability10(7), 2519. Additionally, when the literature review is expanded, the discussion section should also refer back to the existing literature, thereby clarifying what is the addition of your own study to the already existing body of knowledge (you already do this to some extent in the conclusion section, but it could be made more specific). the main issue I have with the paper is that it needs extensive English editing. Many errors remain regarding spelling and grammar and it is recommended to have it proofread by a native English speaker.

Reviewer 4 Report

Review of the paper „A New Extension to Multi-Criteria Decision-Making model for Sustainable Supplier Selection under Intuitionistic Fuzzy Environment”

I would like to thank the editor for the invitation to review this paper.

Generally I am satisfied of the quality and the paper topic and contents.

Some detailed comments:

Abstract: quite informative, well written

Introduction: quite good motivation of the study

Literature review: enough wide, sources are properly chosen and well commented, however authors can add some more paper on Sustainable Supply Chain Management from Sustainability journal

Methodology: well presented, quite sophisticated

The Case Study is very valuable and professional

Unfortunately there is luck of results discussion in relation to previous studies. Authors should evaluate their approach and methodology

Round 2

Reviewer 2 Report

Thank you for the revision – the proofreading and editing has helped improve the clarity of communication and made it easier to follow/understand the research. The new additions, while relatively minor, have also helped to explain more clearly the contribution of the research and how and why this is important. In the present form, the manuscript will be much more useful to our research community. My research team and I would be citing this type of research increasingly.

A final proofread and check on spelling/grammar is recommended as there are still a few small errors. For example: Line 559: “chain, This result were compatible”

Author Response

Dear Reviewer 2,

      First of all, we are very grateful to the reviewer 2 for your valuable comments and suggestions that have much improved the manuscript and have a guideline for our further paper. In the revised manuscript, according to the recommendation for a final proofread and check on spelling/grammar, we have carefully addressed these comments and suggestions, and these revisions are marked in red color for grammar check and for spelling errors check (such as space, comma, and full stop) are marked in blue color. The following is shown in the main revisions, and we have attached a file for the manuscript as below.

Thank you for your consideration 

Best regards, 

Author 

Reviewer 3 Report

Dear authors,

Thank you for revising your manuscript according to the reviewer comments. I think you have addressed the comments in a detailed manner and my recommendation is to accept this paper for publication, after careful English proofreading (E.g. the last sentence of the abstract).

Author Response

Dear Reviewer 3,

     First of all, we are very grateful to the reviewer 3 for your valuable comments and suggestions that have much improved the manuscript and have a guideline for our further paper. In the revised manuscript, according to the recommendation for a final proofread and check on spelling/grammar, we have carefully addressed these comments and suggestions, and these revisions are marked in red color for grammar check and for spelling errors check (such as space, comma, and full stop) are marked in blue color. The following is shown in the main revisions, and we have attached a file for the manuscript as below.

Thank you for your consideration 

Best & regard

Author 

This manuscript is a resubmission of an earlier submission. The following is a list of the peer review reports and author responses from that submission.

Round 1

Reviewer 1 Report

This paper proposes a new decision-making approach for selecting sustainable suppliers. It is clear that authors have done an extensive review of the literature, describing their merits and scopes of improvement. However, in the current format, the paper needs to be significantly improved to make it readable and comprehensible. The major drawback of this paper is far too many acronyms used which makes the paper confusing, complicated to read and almost impossible to understand for readers. 

The authors may want to use acronyms just for the main variables, i.e., SSS, SSCM, and MCDM. However, currently, the paper also uses GSCM, DEMATEL, ANP, VIKOR,  IF-DANP-mV, TOPSIS, PROMETHEE, ELECTRE, INRM, GDEA, DANP, ........ (the list continues). These are are just way too many for readers to follow through, throughout the length of the paper. Also, do we really need to refer to six or more frameworks, i.e., GSCM, MCDM, DEMATEL, ANP, IFS, and VIKOR, to achieve the goal of this paper? A model should be parsimonious and be able to provide the simplest viable explanation of the phenomenon.

The research gap of this paper (for example in #139-140) is not very convincing. A more in-depth analysis is a subjective perception, which needs more factual support in order to be determined as a research gap.

Also, in the methodology, authors missed out providing textual interpretation and description of their mathematical framework and derivations. As a reader, I am interested in not just how it is done, but also why it is done and what it means in the context of the paper. 

Use of only one case-study to demonstrate and evaluate such a convoluted and sensitive approach is not sufficient. 

Authors may want to rethink about who their readers and audiences are. Often, the industry persons are unfamiliar with such complicated terms, and hence they might not be able to utilize any information from this paper to their advantage. Academic scholars, who are not directly associated with this concept, might find it challenging to connect all the multiple frameworks in order to understand the derived model. It seems that only the academic scholars, that too only those who work specifically on this topic, would be able to potentially comprehend and benefit from this paper. Unfortunately, with such a limited audience, the contribution of this paper is questionable. 

To improve this paper, authors may want to revisit and rewrite the whole paper, using a more simplified approach. It seemed that many times authors assumed that their readers would be familiar with all the terms and their definitions, and hence, they skipped describing the core terms. Authors need to decide what exactly they are trying to say and just stick to describing that. Reduced used of acronyms, describing all the important terms, hashing out unnecessary terms and author names, offering textual interpretation for the mathematical components, can be some examples of the simplified approach. Additionally, missing words, missing components of alphanumeric codes (eg., 31 instead of C31), mistaken grammar, complex and long sentences, incomplete sentences make it difficult to read. Sentences need to be short, succinct, and to the point. The paper needs extensive editing of the English language, including grammatical errors and sentence constructions. 

Reviewer 2 Report

The paper entitled "A New Extension to Multi-Criteria Decision-Making 3 model for Sustainable Supplier Selection under Intuitionistic Fuzzy Environment" focuses on the new extension to multi-criteria decision making model (MCDM) under intuitionistic fuzzy environment for sustainable supplier selection (SSS) based on SSCM practices. The paper employs a case study of sustainable supplier selection in Thailand palm oil products industry to illustrate its approach and methods.

1) The Abstract should contain the concise results of the paper and explain its main outcomes in brief.

2) The Literature review should be extended. Currently, the paper has 45 sources which is not sufficient. At least 20-30 more relevant sources should be added.  

3) Section 4.2 "Proposed criteria for SSS based SSCM practices" should be elaborated to explain how the respondents were selected and recruited. What about the questionnaires? How were those prepared and administered? The section states that interviews with 6 managers were held - this is a very small sample that hinders the validity of the results. It should be explained that the results have to be regarded with care due to the limitation of data.

4) The Conclusions need to be ameliorated and expanded in order to explain the outcomes and implications of the paper. It cannot be just a repetition of results that were presented in the previous sections.

Overall, the paper is robust and scientifically sound. It also reads well. The paper can be published after the comments and suggestions presented above are tackled by the authors.

Reviewer 3 Report

Dear author(s) – thank you for providing an interesting manuscript for review. This is an important topic that has significant impact for both practice and theory. Overall, the manuscript is well-written and carefully constructed but there are several areas where I think improvements should be made.

1. Introduction – can you be more specific about what the real-world problem is or what is motivating the study? You seem to be getting close to this point on p. 2, lines 45-58 but you seem to leave this at the level of ‘no research in this area’ or ‘no research on this element of the problem’. While that may be the case (and I think you demonstrate this clearly in the literature review section), can you be more specific about why this is a problem in practice or a problem in theory? Has the inability to select suppliers in this way been cited by professionals? Is there a gap where (for example).

Therefore, I think that the motivation within the introduction can be better articulated. Why should a reader care about this issue? You might use professionally focused literature in addition to the academic literature you cite. At the moment – this research fails the ‘so what’ test for me.

A scholar, whose work I admire, is Craig R Carter – and I believe he tends to do an exceptional job at motivating the study. Generally, I find that he is able to articulate the importance of his research. If you read some of his papers, you generally see that he has worked through – quite regularly – non-academic literature, showing why this is an important issue that needs to be solved.

Example: see Carter & Rogers (2008). Second paragraph – use of KPMG report to establish the practical importance of this topic. Later, they note that “[t]he answers to these research questions will help to clarify and begin to defuse the debate surrounding the relationship between environmental and social performance on one hand, and economic performance on the other.” This positions their research and drives home the importance.

Singhal is often very explicit in the way he addresses the issue of importance and contribute. E.g., Jacobs & Singhal (2013), read paragraphs 2-5 where they outline very carefully and explicitly what they are doing.

See Hendricks & Singhal (2005) on p. 36: “Our analysis of the long-run stock price effects of disruptions is important for a number of reasons.” Then, they carefully explain the managerial importance and implications for practice over this and the following paragraph. The first paragraph is a logical argument while the second paragraph is supported by references.

Over these examples, note how Carter & Rogers used a professional (KMPG) report, Hendriks & Singhal use both logical arguments and discussions grounded in other academic literature.

2. It is good that you were clear about the objectives/aims of the research (p.2). Can you come back to these aims at the end of the manuscript and more clearly explain how you have met them or addressed these aims? 

*** Literature review

This section is carefully written and addresses the elements that I expect to see. Table 1 is helpful – Can you update the table to include specific citations?

The end of the literature review notes: “To fill this limitation, this study build IF-DANP-mV model aims to enhanced decision making under fuzzy environment and investigate for more in-depth analysis the interrelationships among criteria.” --- are you able to better connect this statement to your introduction and conclusion to strengthen the logic in the article?

*** Methods.

This section appears to be carefully detailed and reads well.

*** Case study & results

This is a good section and I am pleased it has been included.

Can you explain why in Section 4.5 the order of alternatives (Table 10) are exactly the same? If this is the case, can you more clearly explain the value of the MCDM approach you develop and propose??? Can you illustrate in what way this is different/superior to other approaches? What is the benefit if it just gives us the same practical solution/ranking of suppliers?

If your proposed approach gives us the same outcome, can you focus on how or why it is a valuable approach.

On p. 19, lines 544 to 549, you note TOPSIS model from Memari et al [35] (note the error where you write “etal.” – there are many small errors in the manuscript like this that must be corrected). Is this a better comparison with your approach?

Are you better able to note the novelty or what is distinct about this research with comparison to several other articles? A ‘compare and contrast’ with a few other key articles may help you achieve this and better highlight the contribution of this article. While you list the contributions in the ‘conclusion’ section this is not with reference to any past research so it is not clear how you have addressed deficiencies in a particular set of research. With no specific articles referenced it is hard to evaluate or judge the novelty of the research and the value of the research you have completed.

***Conclusions

Can you explicitly provide some areas for future research?

REFERENCES

Carter, C. R. & Rogers, D. S. (2008). A framework of sustainable supply chain management: moving toward new theory, International Journal of Physical Distribution & Logistics Management, 38(5), 360-387

Hendricks, B. and Singhal, V. R. (2005). Effect of supply chain disruptions on long-run stock price performance and equity risk of the firm, Production and Operations Management, 14(1), 35-52.

Jacobs, B. & Singhal, V. R. (2013). The effect of product development restructuring on shareholder value, Production and Operations Management, 23(5), 728-743.